# LLM-Driven Policy Diffusion: Enhancing Generalization in Offline Reinforcement Learning

## Abstract

Reinforcement Learning (RL) is known for its strong decision-making capabilities and has been widely applied in various real-world scenarios. However, with the increasing availability of offline datasets and the lack of well-designed online environments from human experts, the challenge of generalization in offline RL has become more prominent. Due to the limitations of offline data, RL agents trained solely on collected experiences often struggle to generalize to new tasks or environments. To address this challenge, we propose LLM-Driven Policy Diffusion (LLMDPD), a novel approach that enhances generalization in offline RL using task-specific prompts. Our method incorporates both text-based task descriptions and trajectory prompts to guide policy learning. We leverage a large language model (LLM) to process text-based prompts, utilizing its natural language understanding and extensive knowledge base to provide rich task-relevant context. Simultaneously, we encode trajectory prompts using a transformer model, capturing structured behavioral patterns within the underlying transition dynamics. These prompts serve as conditional inputs to a context-aware policy-level diffusion model, enabling the RL agent to generalize effectively to unseen tasks. Our experimental results demonstrate that LLMDPD outperforms state-of-the-art offline RL methods on unseen tasks, highlighting its effectiveness in improving generalization and adaptability in diverse settings.

## 1 Introduction

Reinforcement Learning (RL) has emerged as a powerful paradigm for sequential decision-making and control, achieving remarkable success across a wide range of applications, from robotics (Tang et al., 2024) and autonomous driving (Lee et al., 2024) to finance (Liu et al., 2022) and healthcare (Yu et al., 2021). With the growing application of RL in real-world scenarios, the limitations of real-world RL environments have become increasingly evident—in many cases, access to either data or the environment is restricted. This highlights the importance of generalization in RL, which focuses on training RL agents on limited datasets or environments while ensuring their generalizability to unseen tasks or environments (Kirk et al., 2023), thereby reducing the need for extensive task-specific training and direct access to all possible environments. This not only enhances efficiency but also significantly reduces human effort in designing and collecting training data.

Generalization in RL however poses significant challenges, as it requires maximizing the performance of RL agents on unseen tasks or environments that were not covered in the training data. Prior research has identified several factors contributing to the generalization gap in deep RL agents, such as overfitting and memorization in deep neural networks, which can lead to poor adaptability (Arpit et al., 2017). To address these challenges, various methods have been explored, including data augmentation to enhance generalization to unseen states (Yarats et al., 2021; Raileanu et al., 2021; Zhang and Guo, 2021), generation of synthetic environments to increase training diversity (Wang et al., 2019; 2020), approaches to reduce discrepancies between different environments (Liu et al., 2020), and optimization strategies that account for environment variations (Raileanu and Fergus, 2021). However, in many real-world applications, direct access to online environments is often impractical. For example, in autonomous driving (Lee et al., 2024), training data is primarily collected from human drivers, resulting in large offline datasets rather than interactive online environments.

This makes generalization in offline RL an especially important area of study. The goal is to train RL agents from offline data that can achieve strong performance while generalizing to unseen circumstances. Prior research has shown that generalization in offline RL is more challenging than in online RL (Mazoure et al., 2022; Mediratta et al., 2024), highlighting both its significance and inherent difficulties.

The main challenges of generalization in offline RL fall into two primary categories: (1) the generalization gap inherited from standard deep RL (Arpit et al., 2017), and (2) the lack of sufficient exploration data in offline RL training. Previous works have employed techniques such as data augmentation (Laskin et al., 2020; Sinha et al., 2022; Modhe et al., 2023) and adversarial training (Qiao and Yang, 2024) to address the generalization gap caused by overfitting. To improve training efficiency in offline RL—where arbitrary exploration like in online RL is not possible—researchers have sought to enhance data utilization (He et al., 2023) and mitigate the effects of out-of-distribution data, thereby improving generalization (Ma et al., 2024; Wang et al., 2024a). However, most prior work has primarily focused on reducing reliance on training data to enhance generalization. Few studies have leveraged readily available task-specific information such as text descriptions from offline data or incorporated easily collectible task-related data to enhance generalization.

In this work, we introduce LLM-Driven Policy Diffusion (LLMDPD), a novel approach to improving generalization in offline RL by leveraging task-specific prompts. We introduce two types of prompts: a text prompt, which is a textual description of the task or environment, and a trajectory prompt, which consists of a single trajectory collected from the target task or environment, both of which are easy and cheap to obtain. First, leveraging the capabilities of large language models (LLMs) in natural language processing (Qin et al., 2024) and knowledge distillation (Xu et al., 2024; Yang et al., 2024), we use a pre-trained LLM to process the text prompt, extracting useful insights from the task description while also drawing on the pre-collected knowledge embedded in the LLM. Second, we train a transformer model to process the trajectory prompt, capturing task-specific behavioral patterns from the transition dynamics of the prompt. Both prompts are encoded into latent embeddings, which serve as conditional inputs to support adaptive and context-aware policy training. We adopt policy diffusion as our policy function, which takes the state and the task-specific prompt embeddings as inputs and outputs a task-aware action distribution, enabling generalization to unseen tasks without fine-tuning. We evaluate LLMDPD on several benchmarks, and our experimental results show that LLMDPD outperforms state-of-the-art methods in offline RL generalization, demonstrating the effectiveness of our approach.

Our key contributions are summarized as follows:

- We are the first to incorporate text prompts leveraging the advantages of LLMs, along with trajectory prompts, into offline RL. This approach significantly enhances the generalizability of offline RL agents to unseen tasks.

- We introduce Context-aware Conditional Policy Diffusion (CCPD), which incorporates prompt information as a conditioning mechanism within the offline RL framework. This approach enhances policy flexibility while improving generalization to unseen tasks.

- Our experimental results show that LLMDPD significantly outperforms state-of-the-art offline RL methods in generalization, demonstrating the effectiveness of our approach.

## 2 RELATED WORKS

**Generalization in Offline RL**   Generalization in RL focuses on addressing the generalization gap to enhance RL agent's ability to perform well on unseen tasks or environments. Traditional generalization studies in RL primarily examine the generalizability of RL agents in online environments. However, with the increasing availability of large offline datasets and the lack of direct access to online environments in many real applications, present research has shifted toward a more challenging yet practical objective: improving generalization in offline RL. Mazoure et al. (2022) systematically analyzed the differences in generalization between online and offline RL, providing theoretical evidence that online RL algorithms struggle to generalize in offline settings. Mediratta et al. (2024) conducted experiments evaluating the generalization capabilities of widely used RL methods in both online and offline settings. Their results indicate that standard RL methods generalize more poorly in offline environments, reinforcing that generalization in offline RL is a more diffi-

cult problem. Laskin et al. (2020) and Sinha et al. (2022) introduced data augmentation schemes to enhance the generalization ability of offline RL agents. He et al. (2023) proposed the Multi-Task Diffusion Model (MTDIFF), which leverages knowledge from multi-task data to improve generalization in offline RL through shared information. Qiao and Yang (2024) introduced Soft Adversarial Offline Reinforcement Learning (SAORL), which imposes constraints on traditional adversarial examples, formulating a worst-case optimization problem to generate soft adversarial examples. Zhao et al. (2024) proposed Offline Trajectory Generalization through World Transformers for Offline Reinforcement Learning (OTTO), a method designed to learn state dynamics and reward functions, thereby enhancing generalization to unseen states. Ma et al. (2024) developed Representation Distinction (RD), a plugin method that improves offline RL generalization by detecting and preventing out-of-distribution state-action pairs. Similarly, Wang et al. (2024a) introduced Adversarial Data Splitting (ADS) to relax rigid out-of-distribution boundaries, ultimately improving generalization in offline RL. Xu et al. (2022) introduced Prompt-based Decision Transformer (Prompt-DT), which adopts a prompt framework to enable few-shot adaptation in offline RL, thereby facilitating few-shot policy generalization. Li et al. (2024) and Wang et al. (2024b) incorporate contextual information into offline meta-RL, offering insights into how context can be utilized to improve task conditioning or policy adaptation. Li et al. (2024) introduced the UNICORN, which provides a theoretical perspective on the importance of task variables and their latent representations by incorporating contextual information. Wang et al. (2024b) proposed the Meta Decision Transformer (Meta-DT), which learns a context-aware world model to encode task-specific information. While their work does not explicitly focus on generalization, it provides a valuable perspective on leveraging contextual cues within offline RL.

**Diffusion-based RL** Diffusion models have recently emerged as a powerful generative modeling approach for capturing complex data distributions, and their application to RL has gained traction. Diffuser (Janner et al., 2022) introduces the concept of using diffusion models to model trajectory distributions in offline RL. Several diffusion-based approaches have since extended this idea. Decision Diffuser (Ajay et al., 2022) conditions trajectory generation on high-level task information, such as rewards. PlanDiffuser (Sharan et al., 2024) integrates diffusion models with planning techniques to enhance precision in control tasks. MetaDiffuser (Ni et al., 2023) learns a contextual representation of tasks as conditional input to the diffusion model, enabling the generation of task-oriented trajectories. Similarly, Hierarchical Diffuser (Chen et al., 2024) decomposes long planning horizons into smaller segments, learning subgoals for each to improve planning efficiency. In addition to trajectory-based diffusion models, Wang et al. (2023) introduced Diffusion Policy to offline RL at the action level rather than the trajectory level, providing greater flexibility and more accurate transitions. Chi et al. (2023) further extended Diffusion Policy to broader RL scenarios, enabling effective policy learning for high-dimensional control tasks. Our proposed work is the first that exploits diffusion policy for generalization in offline RL.

**Applications of LLM in RL** Large Language Models (LLMs) have demonstrated remarkable capabilities in understanding, generating, and reasoning over text, making them powerful tools for a wide range of applications. Recently, many studies have started exploring the integration of LLMs into RL to enhance learning efficiency and decision-making. Sun et al. (2024) surveyed the application of LLMs in multi-agent RL frameworks, highlighting key advancements and future research directions. Cao et al. (2024) reviewed the existing literature on LLM-enhanced RL, summarizing advancements and challenges in the field. Du et al. (2023) introduced Exploring with LLMs (ELLM), a method that guides RL pretraining by generating prompted descriptions of the agent's current state. Wang et al. (2024c) proposed LLM-Empowered State Representation (LESR), which leverages LLMs to generate task-relevant state representations, thereby improving the training efficiency of standard RL methods. Shi et al. (2024) introduced Language Models for Motion Control (LaMo), a framework that leverages pretrained language models to enhance Decision Transformers (Chen et al., 2021) and improve the quality of policies learned from offline RL. Yang and Xu (2024) proposed the Language Model-initialized Prompt Decision Transformer (LPDT), which builds upon the standard Prompt-DT framework by replacing the learned prompt embeddings with those derived from a pretrained language model. They further fine-tune the model using Low-Rank Adaptation (LoRA) (Hu et al., 2022) to improve performance over Prompt-DT. Although LLMs have been applied across various domains of RL, their potential for improving generalization in RL remains largely unexplored. Our proposed work is the first to exploit the capacity of LLMs to enhance generalization in offline RL.

```
Task: Meta-World push.
Objective: Push the puck to a goal.
Constraints: Randomize puck and goal positions.
...
```

Figure 1: An example of a text prompt, where the description is structured as a short instruction guiding the LLM on how to refine the prompt.

## 3 METHOD

**Problem setting**   In generalization learning for offline RL, we assume an offline dataset $\mathcal{D} = \{(\mathcal{D}^i, z_{\text{text}}^i, z_\tau^i)\}_{i=1}^m$ that contains data for $m$ seen tasks is given, where the training data $\mathcal{D}^i$ for each task $i$ contains a collection of offline trajectory instances, paired with two additional prompts: a text prompt $z_{\text{text}}^i$ and a trajectory instance prompt $z_\tau^i$. The text prompt provides a textual description for the corresponding task, while the trajectory prompt consists of a *single* trajectory collected from the given task using a behavior policy. Both prompts are easily collectible for either training or test tasks and can provide task-specific context information for the RL model, thereby supporting generalization to unseen tasks. Our goal is to learn an optimal policy $\pi^\star$ from the offline dataset $\mathcal{D}$ over multiple seen tasks such that it can generalize effectively to unseen tasks without fine-tuning, guided by the text and trajectory prompts from the target unseen tasks.

In this section, we present LLM-Driven Policy Diffusion (LLMDPD), a novel approach to enhancing generalization in offline RL. LLMDPD leverages both text and trajectory prompts for adaptive policy diffusion learning. We utilize a pre-trained large language model (LLM) and a parametric transformer as the embedding module to encode the text prompt and trajectory prompt, respectively. The resulting prompt embeddings capture task specific information and serve as conditional inputs for context-aware policy diffusion. We use diffusion Q-learning to jointly train the prompt embedding module and the policy diffusion module in an end-to-end manner, inducing prompt-based policy functions with enhanced adaptability and generalizability. The approach is elaborated below.

### 3.1 PROMPT EMBEDDING

#### 3.1.1 LLM-DRIVEN TEXT PROMPT EMBEDDING

The text prompt $z_{\text{text}}$ for each task consists a natural language description that provides explicit information for the corresponding task. It can be utilized to extract high-level semantic representations of the task, supporting subsequent learning. To facilitate information extraction, we convert the text descriptions of the tasks into a structured format for expressing information of various components, including the task name, objective, constraints, and other specific attributes. An example of the structured text prompt $z_{\text{text}}$ is provided in Figure 1.

Next we utilize a pre-trained large language model (LLM), denoted as $\mathcal{M}$, to produce a latent prompt embedding $\mathbf{z}_{\text{text}}$ from the structure text prompt $z_{\text{text}}$. By harnessing LLMs' ability to process natural language texts and leveraging knowledge distillation from their embedded prior knowledge, the latent prompt embedding is expected to encode rich task-relevant information. To enhance the efficiency of prompt interpretation, we also include a default brief instruction, e.g., "convert the following task description into a structured policy representation", to guide the LLM in processing the text prompt. Specifically, by using the structured text prompt together with the interpretation instruction as input, we obtain an embedding output by performing mean pooling to the token embeddings produced from the last hidden layer of the LLM, effectively capturing the overall context to ensure a comprehensive representation of the processed prompt. To enable adaptation to the subsequent policy learning task, we further introduce a multilayer perceptron (MLP) project head $h_\psi$ parameterized by $\psi$ on top of the LLM $\mathcal{M}$, refining the embedding output to a final embedding vector $\mathbf{z}_{\text{text}}$. This embedding process can be expressed using the following equation:

$$\mathbf{z}_{\text{text}} = h_\psi(\mathcal{M}(z_{\text{text}})). \tag{1}$$

Here the default interpretation instruction is omitted for simplicity. The parametric projection head can be trained end-to-end within the overall policy learning framework.

### 3.1.2 TRANSFORMER-DRIVEN TRAJECTORY PROMPT EMBEDDING

The trajectory instance prompt consists of a single trajectory collected from the corresponding task, represented as a sequence of state-action transitions, such as

$$z_\tau = [\mathbf{s}_0, \mathbf{a}_0, \mathbf{s}_1, \mathbf{a}_1, \cdots, \mathbf{s}_t, \mathbf{a}_t, \cdots, \mathbf{s}_T, \mathbf{a}_T] \tag{2}$$

where $\mathbf{s}_t$ and $\mathbf{a}_t$ denote the state and action at timestep $t$ respectively, and $T$ denotes the length of the trajectory prompt. Unlike text prompts which explicitly provide task-specific descriptions, a trajectory prompt captures the transition dynamics and behavior patterns of the environment for the corresponding task. To generate informative embeddings from the trajectory prompts, we devise a parametric transformer as the encoder for the trajectory prompts, leveraging Transformer's ability for capturing long-range dependencies in sequential data and supporting effective structural information extraction (Vaswani et al., 2017). Similar to text prompt embedding, we apply mean pooling over the transformer's output and deploy a parametric MLP projection head on top of it. We use a function $g_\varphi$, parameterized by $\varphi$, to denote the overall trajectory prompt encoder that includes both the transformer and the MLP projection head. The final prompt embedding $\mathbf{z}_\tau$ for a trajectory prompt $z_\tau$ can be produced from the encoder $g_\varphi$ as follows:

$$\mathbf{z}_\tau^i = g_\varphi(z_\tau). \tag{3}$$

This transformer based encoder $g_\varphi$ can be trained in an end-to-end manner through back-propagation within the policy learning framework, ensuring that the prompt embedding effectively supports the learning of a context-aware adaptive policy.

### 3.2 CONTEXT-AWARE CONDITIONAL POLICY DIFFUSION

To effectively leverage prompt embeddings that encode rich task-specific information to support generalizable policy learning from offline data, we propose to learn a context-aware conditional policy diffusion (CCPD) module as our policy function: $\pi_\theta(\mathbf{a}|\mathbf{s}, \mathbf{z}_\text{text}, \mathbf{z}_\tau)$, where $\theta$ denotes the parameters of the module. This function conditions the policy generation on task-specific contexts encoded by the text prompt embedding $\mathbf{z}_\text{text}$ and trajectory prompt embedding $\mathbf{z}_\tau$.

The diffusion process of the CCPD module consists of two Markov Chain processes: a forward process and a reverse process. The forward process incrementally adds noise to an action $\mathbf{a}^0$ sampled from offline data, transforming it into a Gaussian prior over $K$ diffusion steps. In the reverse process, starting from a Gaussian noise prior $\mathbf{a}^K \sim \mathcal{N}(\mathbf{0}, \boldsymbol{I})$, the model progressively denoises the action at each timestep $k$, conditioned on the given state $\mathbf{s}$ and the corresponding prompt embeddings $\mathbf{z}_\text{text}$ and $\mathbf{z}_\tau$. Specifically, at timestep $k$, the next action $\mathbf{a}^{k-1}$ in the sequential denoising process is generated from the following Gaussian distribution:

$$p_\theta(\mathbf{a}^{k-1}|\mathbf{a}^k, \mathbf{s}, \mathbf{z}_\text{text}, \mathbf{z}_\tau) = \mathcal{N}(\mathbf{a}^{k-1}; \mu_\theta(\mathbf{a}^k, \mathbf{s}, \mathbf{z}_\text{text}, \mathbf{z}_\tau, k), \sigma_k^2 \mathbf{I}) \tag{4}$$

$$\text{with } \mu_\theta(\mathbf{a}^k, \mathbf{s}, \mathbf{z}_\text{text}, \mathbf{z}_\tau, k) = \frac{1}{\sqrt{\alpha_k}} \left( \mathbf{a}^k - \frac{1 - \alpha_k}{\sqrt{1 - \bar{\alpha}_k}} \epsilon_\theta(\mathbf{a}^k, \mathbf{s}, \mathbf{z}_\text{text}, \mathbf{z}_\tau, k) \right)$$

Here, $\bar{\alpha}_k = \prod_{i=1}^k \alpha_i$, and $\alpha_k$ follows a predefined variance schedule (Ho et al., 2020); $\epsilon_\theta$ denotes a learned noise prediction network which estimates the added noise at each diffusion step, allowing the model to recover the clean action after $K$ timesteps. This CCPD module is trained together with the prompt embedding module on the offline data $\mathcal{D}$ by minimizing a diffusion loss $\mathcal{L}_d(\psi, \varphi, \theta)$, defined as the mean squared error (MSE) between the true noise $\epsilon$ and the predicted noise:

$$\mathcal{L}_d(\psi, \varphi, \theta) = \mathbb{E}_\mathcal{C} \left[ \left\| \epsilon - \epsilon_\theta \left( \sqrt{\bar{\alpha}_k} \mathbf{a} + \sqrt{1 - \bar{\alpha}_k} \epsilon, \mathbf{s}, h_\psi(\mathcal{M}(z_\text{text}^i)), g_\varphi(z_\tau^i), k \right) \right\|^2 \right] \tag{5}$$

$$\text{where } \mathcal{C} = \{i \sim [1 : m], (\mathbf{s}, \mathbf{a}) \sim \mathcal{D}^i, k \sim [1 : K], \epsilon \sim \mathcal{N}(\mathbf{0}, \boldsymbol{I})\}.$$

Given a trained policy diffusion module, the policy function $\pi_\theta$ is obtained by progressively denoising from a Gaussian prior, following the reverse diffusion process indicated by Eq.(4).

**Incorporating Reward Maximization via Actor-Critic Policy Diffusion** Minimizing only the diffusion loss $\mathcal{L}_d(\psi, \varphi, \theta)$ results in a behavior-cloned policy, which mimics the offline dataset $\mathcal{D}$ without optimizing for rewards. To address this problem, we introduce a Q-function $Q_\phi(\mathbf{s}, \mathbf{a})$ to

estimate the expected cumulative reward. Specifically, we deploy the double Q-Learning strategy (Hasselt, 2010) that uses two Q-networks, $Q_{\phi_1}$ and $Q_{\phi_2}$, to prevent Q-value overestimation, which are trained by minimizing the following Q-losses (Wang et al., 2023):

$$\mathcal{L}_q(\phi_\ell) = \mathbb{E}_{\mathcal{C}_q}\left[\left\|\left(r_t + \gamma \min_{\ell'=1,2} Q_{\bar{\phi}_{\ell'}}(\mathbf{s}_{t+1}, \mathbf{a}_{t+1}^0)\right) - Q_{\phi_\ell}(\mathbf{s}_t, \mathbf{a}_t)\right\|^2\right], \quad \text{for } \ell \in \{1, 2\} \quad (6)$$

$$\text{where } \mathcal{C}_q = \{i \sim [1:m], (\mathbf{s}_t, \mathbf{a}_t, r_t, \mathbf{s}_{t+1}) \sim \mathcal{D}^i, \mathbf{a}_{t+1}^0 \sim \pi_\theta(\cdot|\mathbf{s}_{t+1}, \mathbf{z}_\text{text}^i, \mathbf{z}_\tau^i)\}$$

where $\bar{\phi}_{\ell'}$ indicates the stopping-gradient network used for Q-value estimation, and $r_t$ denotes the reward observed in the trajectories of the offline dataset. The two Q-networks will converge to the same solution $\phi$ in the limit. We utilize the Q-values estimated by either one of the Q-networks ($\phi = \phi_1$ or $\phi = \phi_2$) to guide the reverse policy diffusion process, generating actions that maximize the expected cumulative reward:

$$\mathcal{L}_r(\psi, \varphi, \theta) = \mathbb{E}_{i \sim [1:m], \mathbf{s} \sim \mathcal{D}^i, \mathbf{a}^0 \sim \pi_\theta(\cdot|\mathbf{s}, h_\psi(\mathcal{M}(z_\text{text}^i)), g_\varphi(z_\tau^i))}\left[Q_\phi(\mathbf{s}, \mathbf{a}^0)\right] \quad (7)$$

To balance behavior cloning and reward maximization, the total training loss for the policy diffusion module is formulated as a weighted combination of $\mathcal{L}_d$ and the negation of the reward objective $\mathcal{L}_r$:

$$\mathcal{L}(\psi, \varphi, \theta) = \mathcal{L}_d(\psi, \varphi, \theta) - \lambda \mathcal{L}_r(\psi, \varphi, \theta) \quad (8)$$

where $\lambda$ is a hyperparameter controlling the trade-off between action denoising and reward-driven optimization. The policy diffusion module can be viewed as an actor and the Q-networks can be treated as critics. They can be simultaneously learned using the actor-critic learning strategy. The overall actor-critic diffusion training algorithm is illustrated in Algorithm 1. By combining policy diffusion with Q-learning, our LLMDPD model learns a generalizable and reward-maximizing policy, capable of adapting to unseen tasks under the guidance of task-aware prompts.

---

**Algorithm 1** LLMDPD Training

---

**Input:** offline dataset $\mathcal{D} = \{\mathcal{D}^i\}_{i=1}^m$, initialized embedding module $(\psi, \varphi)$ with pre-trained LLM $\mathcal{M}$, initialized policy diffusion module $\theta$, initialized Q-networks $Q_{\phi_1}$ and $Q_{\phi_2}$
**Output:** Trained model parameters $\psi, \varphi, \theta, \phi_1, \phi_2$.
1: **for** each epoch **do**
2:     Sample a task $i \sim [1:m]$.
3:     Extract text prompt $z_\text{text}^i$ and trajectory prompt $z_\tau^i$ for task $i$.
4:     Sample a batch $\mathcal{B} = \{(\mathbf{s}_t, \mathbf{a}_t, r_t, \mathbf{s}_{t+1}), \cdots\}$ from seen offline data $\mathcal{D}^i$.
5:     **for** each transition $(\mathbf{s}_t, \mathbf{a}_t, r_t, \mathbf{s}_{t+1})$ in batch $\mathcal{B}$ **do**
6:         Compute prompt embeddings: $\mathbf{z}_\text{text}^i = h_\psi(\mathcal{M}(z_\text{text}^i))$ and $\mathbf{z}_\tau^i = g_\varphi(z_\tau^i)$.
7:         Sample action: $\mathbf{a}_{t+1}^0 \sim \pi_\theta(\cdot|\mathbf{s}_{t+1}, \mathbf{z}_\text{text}^i, \mathbf{z}_\tau^i)$.
8:         Update the Q-networks $\phi_1, \phi_2$ by minimizing the Q-loss in Eq.equation 6
9:         Randomly select $\phi_1$ or $\phi_2$ as the critic $\phi$
10:         Sample action: $\mathbf{a}_t^0 \sim \pi_\theta(\cdot|\mathbf{s}_t, \mathbf{z}_\text{text}^i, \mathbf{z}_\tau^i)$.
11:         Sample diffusion timestep $k \sim [1:K]$ and noise $\epsilon \sim \mathcal{N}(\mathbf{0}, \mathbf{I})$.
12:         Update parameters $\psi, \varphi, \theta$ by minimizing the total loss in Eq.equation 8.
13:     **end for**
14: **end for**

---

## 4  EXPERIMENTS

To thoroughly evaluate the generalization performance of our LLMDPD method, we conducted experiments on the Meta-World dataset (Yu et al., 2020) and the D4RL dataset (Fu et al., 2020), both of which serve as benchmarks for evaluation an RL agent's generalizability to unseen tasks.

### 4.1  EXPERIMENT ON META-WORLD

**Environment**  Meta-World (Yu et al., 2020) is a widely used benchmark designed for multi-task and meta-RL. It is implemented using the MuJoCo physics engine (Todorov et al., 2012), which provides a diverse set of near-realistic robotic manipulation tasks, such as picking, pushing, and

Table 1: This table presents the average success rates for various comparison methods on Meta-World-V2 tasks, evaluated over 500 episodes per task. Results are averaged over three runs.

| Type | Task | SAC | S4RL | RAD | MTDIFF | LLMDPD |
|------|------|-----|------|-----|--------|--------|
| Unseen | box-close | $23.46 \pm 7.11$ | $73.13 \pm 3.51$ | $71.20 \pm 4.84$ | $65.73 \pm 8.36$ | $\mathbf{75.24 \pm 4.90}$ |
| | hand-insert | $30.60 \pm 9.77$ | $60.20 \pm 1.57$ | $43.79 \pm 3.44$ | $70.87 \pm 3.59$ | $\mathbf{78.71 \pm 5.33}$ |
| | bin-picking | $42.13 \pm 14.33$ | $72.20 \pm 4.17$ | $43.27 \pm 4.38$ | $55.73 \pm 7.63$ | $\mathbf{74.65 \pm 6.28}$ |
| Seen | sweep-into | $91.80 \pm 1.14$ | $90.53 \pm 3.52$ | $88.06 \pm 9.86$ | $\mathbf{92.87 \pm 1.11}$ | $91.84 \pm 2.42$ |
| | coffee-push | $28.60 \pm 14.55$ | $28.73 \pm 8.44$ | $33.19 \pm 2.86$ | $74.67 \pm 6.79$ | $\mathbf{76.56 \pm 2.89}$ |
| | disassemble | $60.20 \pm 16.29$ | $52.20 \pm 5.68$ | $60.93 \pm 20.80$ | $69.00 \pm 4.72$ | $\mathbf{72.30 \pm 6.48}$ |

reaching. A notable subvariant, Multi-Task 50 (MT50), consists of 50 robotic manipulation tasks well-suited for offline data collection, with each task accompanied by a detailed description. Among these 50 tasks, 45 are designated as training tasks, while the remaining 5 serve as unseen test tasks to evaluate an RL agent's generalizability to novel tasks. With its pre-collected offline data and comprehensive task descriptions, Meta-World serves as an ideal testbed for our LLMDPD method.

**Comparison Methods** We compare our LLMDPD method against four baselines on the Meta-World benchmark: SAC, S4RL, RAD, and MTDIFF. SAC (Haarnoja et al., 2018) is an off-policy actor-critic RL algorithm used to collect the offline dataset in Meta-World, serving as a fundamental baseline. RAD (Laskin et al., 2020) applies data augmentation in the state space, enhancing generalization, particularly for image-based observations. S4RL (Sinha et al., 2022) extends this idea by integrating advanced state-space augmentation techniques to improve generalization in offline RL. MTDIFF (He et al., 2023) is a diffusion-based multi-task RL method that facilitates implicit knowledge sharing across tasks, enabling better adaptation to unseen tasks.

**Implementation Details** For offline data collection in Meta-World tasks, we follow the same criteria as (He et al., 2023), using SAC (Haarnoja et al., 2018) to pre-collect 40M timesteps of offline data. Our model adopts the same transformer architecture as MTDIFF (He et al., 2023) and learns policy diffusion based on the Diffusion-QL framework (Wang et al., 2023). We use the formal text descriptions from Meta-World (Yu et al., 2020) as text prompts and employ the same trained SAC agent to generate trajectory prompts. We primarily use Llama3-7B (Dubey et al., 2024) as our base LLM, and the last hidden layer embeddings are mean-pooled and then passed through a 3-layer MLP projection head (hidden size 512, ReLU activation) to obtain the final text prompt embedding. The RL agent is trained on three seen tasks: sweep-into, coffee-push, and disassemble, and its generalization performance is evaluated on three unseen tasks: box-close, hand-insert, and bin-picking.

**Experimental Results** The experimental results for Meta-World tasks are presented in Table 1. We evaluate the generalization ability of our LLMDPD method on three unseen tasks while using its performance on seen tasks to evaluate its overall sample efficiency during training. The results show that LLMDPD significantly outperforms all other methods on the three unseen tasks. Notably, on the bin-close task, LLMDPD achieves an 7.84 improvement in average success rate compared to the previous best method, MTDIFF. Similarly, it shows an improvement of 2.45 on bin-picking and 2.45 on hand-insert, demonstrating strong generalization capabilities, even when compared to state-of-the-art data augmentation methods.

On seen tasks, our LLMDPD method achieves strong results, outperforming all other methods on coffee-push and disassemble tasks. This demonstrates that LLMDPD not only exhibits strong generalization on unseen tasks but also efficiently learns from training on seen tasks. However, on the sweep-into task, our method falls slightly behind MTDIFF. This may be because text descriptions primarily enhance performance on complicated tasks, while offering limited gains on simpler tasks.

## 4.2 EXPERIMENT ON D4RL

**Environment** D4RL (Fu et al., 2020) is a benchmark dataset for offline RL, aiming to simulate real-world applications. Its locomotion suite, built on the MuJoCo physics engine (Todorov et al., 2012), includes pre-collected offline datasets at three expertise levels: medium-replay, medium, and medium-expert. D4RL is widely used to evaluate an offline RL agent's generalization ability, as its datasets do not fully cover all possible state-action pairs. Among them, medium-replay consists of

Table 2: This table presents the normalized scores of various comparison methods on the D4RL locomotion suites using medium-replay offline data. Results are averaged over three runs.

| Environment | PnF-Qgrad | SAORL | Diffusion-QL | OTTO | RD | ADS | LLMDPD |
|---|---|---|---|---|---|---|---|
| halfcheetah | $41.2 \pm 4.5$ | $40 \pm 2.6$ | $47.8 \pm 0.3$ | $47.8 \pm 0.2$ | $57.7 \pm 0.9$ | $\mathbf{59.4 \pm 3.1}$ | $57.3 \pm 3.6$ |
| hopper | $51.6 \pm 15.5$ | $68 \pm 8.8$ | $101.3 \pm 0.6$ | $103.8 \pm 0.6$ | $104.1 \pm 0.8$ | $105.0 \pm 0.9$ | $\mathbf{105.9 \pm 1.5}$ |
| walker2d | $70.4 \pm 2.0$ | $80 \pm 23$ | $95.5 \pm 1.5$ | $93.6 \pm 2.2$ | $92.1 \pm 2.7$ | $96.1 \pm 0.6$ | $\mathbf{102.2 \pm 0.8}$ |
| **Average** | 54.4 | 62.7 | 81.5 | 81.7 | 84.6 | 86.8 | **88.5** |

Table 3: This table presents the average success rates for all ablation variants on Meta-World-V2 tasks, evaluated over 500 episodes per task. Results are averaged over three runs.

| Task | LLMDPD | LLMDPD-OLMo-1B | w/o-prompt | w/o-$z_{text}$ | w/o-$z_\tau$ | w/o-DP |
|---|---|---|---|---|---|---|
| box-close | $\mathbf{75.24 \pm 4.90}$ | $72.18 \pm 5.66$ | $67.29 \pm 6.89$ | $70.14 \pm 8.57$ | $72.01 \pm 8.37$ | $70.53 \pm 6.44$ |
| hand-insert | $\mathbf{78.71 \pm 5.33}$ | $76.61 \pm 4.21$ | $69.31 \pm 5.38$ | $72.16 \pm 7.93$ | $75.43 \pm 6.70$ | $73.19 \pm 6.31$ |
| bin-picking | $\mathbf{74.65 \pm 6.28}$ | $71.76 \pm 6.39$ | $61.48 \pm 8.85$ | $64.55 \pm 10.73$ | $70.13 \pm 8.64$ | $66.86 \pm 9.59$ |

all samples collected during training until the policy reaches the medium level. Lacking a clean and optimal behavior policy, it is well-suited for evaluating an RL agent's generalization capability.

**Comparison Methods** We evaluate six comparison methods on the D4RL dataset: PnF-Qgrad, SAORL, Diffusion-QL, OTTO, ADS, and RD. PnF-Qgrad (Modhe et al., 2023) augments unseen states to fine-tune hyperparameters for existing offline RL methods, and we use its COMBO variant, referred to as PnF-Qgrad. SAORL (Qiao and Yang, 2024) enhances generalization by learning soft adversarial examples, while Diffusion-QL (Wang et al., 2023) trains a diffusion policy to maximize offline RL performance. OTTO (Zhao et al., 2024) leverages World Transformers to simulate high-reward trajectories for improved generalization, and we adopt its best variant, CQL+OTTO, referred to as OTTO. ADS (Wang et al., 2024a) splits data and generates adversarially hard examples to enhance generalization, and we use its best variant, MCQ+ADS, referred to as ADS. Finally, RD (Ma et al., 2024) improves generalization by differentiating in-sample and OOD state-action pairs, and we adopt its best variant, TD3-N-UNC+RD, referred to as RD. These methods provide a strong benchmark for evaluating our approach against state-of-the-art offline RL generalization techniques.

**Implementation Details** We adopt the model architecture discussed in the previous section. The text prompt consists of task descriptions derived from the base MuJoCo environments (Todorov et al., 2012), along with detailed representations of action and state observations, the reward function, initial state, and termination conditions. The RL agent is trained on the pre-collected D4RL offline datasets of HalfCheetah, Hopper, and Walker2D at the medium-replay expertise level.

**Experimental Results** The experimental results on the D4RL dataset are presented in Table 2. We evaluate the ability of the offline RL agent to generalize to unseen states and actions compared to the medium-replay offline datasets. The results show that LLMDPD achieves the highest overall performance based on average normalized scores. It also attains the best normalized scores in the Hopper and Walker2D environments, while in HalfCheetah, it performs comparably to the best method, ADS. These results demonstrate that LLMDPD not only generalizes to unseen tasks but also exhibits strong sample efficiency in training on offline datasets and adapting to unseen state observations not covered in the offline data.

### 4.3 ABLATION STUDY

We conducted an ablation study on six variants of our model across three unseen tasks: (1) 'LLMDPD', our full model, which includes all components; (2) 'LLMDPD-OLMo-1B', which replaces the base LLM with OLMo-1B (Groeneveld et al., 2024) for processing text prompt embeddings; (3) 'w/o-prompt', which removes both text and trajectory prompts; (4) 'w/o-$z_{text}$', which excludes only the text prompt; (5) 'w/o-$z_\tau$', which excludes only the trajectory prompt; and (6) 'w/o-DP', which replaces the policy's diffusion model with a standard trajectory-based diffusion model. This study systematically evaluates each component's contribution to overall performance.

The results of the ablation study are presented in Table 3. The full model, 'LLMDPD', achieves the highest performance across all three tasks. Removing any component results in a performance drop,

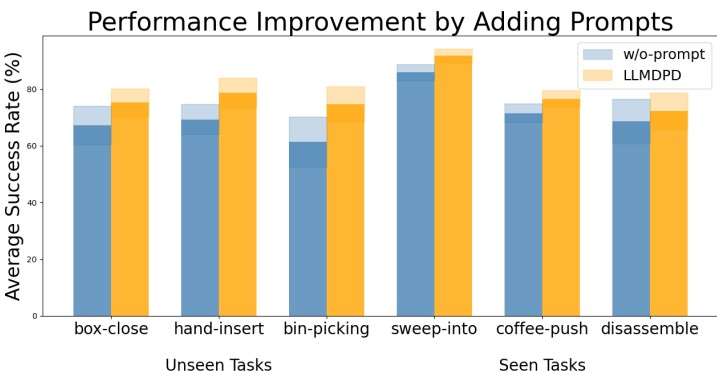

Figure 2: The figure illustrates the performance improvement achieved by incorporating prompts into our LLMDPD method. The blue column represents the average success rate of the 'w/o-prompt' ablation variant, while the orange column represents the full LLMDPD model. The shaded area indicates the standard deviation.

demonstrating the effectiveness of each part of our method. Replacing the base LLM with a smaller model in 'LLMDPD-OLMo-1B' leads to a decline in performance on all three tasks, indicating that larger models provide more detailed guidance to the policy by leveraging natural language processing and pre-collected knowledge. However, this variant still maintains strong generalization performance. Excluding both the text and trajectory prompts ('w/o-prompt') causes a significant performance drop, highlighting the importance of prompting in our method. The 'w/o-prompt' variant, which removes only the text prompt, results in an even more severe decline, particularly on hand-insert and bin-picking tasks. This suggests that without explicit textual guidance, the RL agent struggles to generalize to complex unseen tasks. Similarly, the 'w/o-$z_\tau$' variant, which removes the trajectory prompt, exhibits degraded performance, demonstrating its role in helping the agent capture transition behaviors in unseen tasks. Additionally, 'w/o-DP', which replaces the policy diffusion model with a standard trajectory-based diffusion model, also experiences a significant performance drop. This demonstrates the importance of policy diffusion in reducing overfitting to seen offline data. Overall, the ablation results highlight the contribution of each component to LLMDPD's performance, with prompts playing a particularly crucial role in achieving optimal generalization.

We provide a visualization of the performance improvement achieved by incorporating both text and trajectory prompts into our model, as shown in Figure 2. This figure illustrates the impact of prompts on the model's performance across six tasks, with three unseen tasks (left) and three seen tasks (right). Across all tasks, the inclusion of prompts consistently enhances the average success rate, with the orange bars (LLMDPD) outperforming the blue bars ('w/o-prompt'). Notably, incorporating prompts leads to a greater improvement in the three unseen tasks compared to the seen tasks, demonstrating better generalization performance beyond training on the existing offline dataset. Additionally, the inclusion of prompts slightly reduces the standard deviation, indicating more stable training on the offline dataset and improved consistency in performance. These findings suggest that prompt-driven learning not only boosts success rates but also contributes to more reliable and robust decision-making. Overall, these results highlight the crucial role of prompt-driven guidance in enhancing task understanding, execution, and generalization in offline RL.-

## 5 CONCLUSION

In this work, we propose LLM-Driven Policy Diffusion (LLMDPD), a novel approach to enhancing generalization in offline RL through task-specific prompts. LLMDPD utilizes both easily collectible text-based task descriptions and single trajectory instances as prompts to guide policy learning. To provide rich task-relevant context, LLMDPD leverages LLMs to encode text-based prompts while using a transformer model to encode trajectory prompts. These prompts serve as conditional inputs to a context-aware policy diffusion module, enabling the RL agent to generalize effectively to unseen tasks. By integrating policy diffusion with Q-learning, LLMDPD employs an actor-critic diffusion algorithm to learn a generalizable, reward-maximizing policy. Experimental results on benchmark tasks show that LLMDPD outperforms state-of-the-art offline RL methods in terms of generalization, demonstrating its effectiveness in improving generalizability and adaptability.

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
