# OpenReview forum: "LLM-Driven Policy Diffusion: Enhancing Generalization in Offline Reinforcement Learning"
_ICLR.cc/2026/Conference — Submitted to ICLR 2026_

### Official Review · Reviewer_NvUL · 2025-10-18

**Soundness:** 1
**Presentation:** 2
**Contribution:** 2
**Rating:** 2
**Confidence:** 4

**Summary:**

This paper proposes LLM-Driven Policy Diffusion, a unified framework that integrates large language models with diffusion-based policy learning to achieve generalist decision-making across diverse multimodal tasks.

**Strengths:**

Language-conditioned diffusion provides more transparent and semantically grounded policy generation.

**Weaknesses:**

The joint use of LLM reasoning and diffusion sampling substantially increases training and inference cost.

The model’s performance is sensitive to ambiguous or imprecise task descriptions.

Most experiments are conducted in simulated or small-scale robotic environments.

**Questions:**

1. The experimental section clearly does not correspond to the claims made in the introduction. In the introduction, the authors repeatedly mention generalization under real-world scenarios, but the experimental section only includes experiments in virtual environments, which I believe is inappropriate.
2. In section 3.2, why is the conditional input of the diffusion policy $z_\tau$ instead of $z_\tau^i$?
3. In Equation 4, the authors do not explain what $\sigma_k$ is.
4. In Equation 6, each Q-function update requires using the diffusion policy to generate $a_{t+1}$, so: 1) How many diffusion steps does the diffusion model used in this paper take? 2) What is the time overhead of this algorithm? 3) How do the authors view the problem that the multi-step generation process of the diffusion model may lead to vanishing gradients during backpropagation or cause training instability?
5. On line 292, the authors mention "LLMDPD model learns a generalizable and reward-maximizing policy," but why LLMDPD can produce a generalizable policy, since the LLM merely expresses the task description as a vector representation.
6. typos in Algorithm 1: “Eq.equation 6” and “Eq.equation 8”
7. The paper's description of Table 1 is unclear. I don't know what the specific meaning of "seen" and "unseen" is. Does it mean the model is trained on seen tasks and tested on unseen tasks? If so, then specifically which tasks is it trained on and which tasks is it tested on?
8. Table 2 is more like testing the results of model training and testing under offline dataset conditions, and the compared tasks and methods are too few. I suggest the authors compare multiple diffusion models on the complete D4RL dataset, including representative algorithms such as Diffuser, DD, IDQL, HDMI, AdaptDiffuser, etc.
9. The paper's explanation of how LLM enhances the model's generalization is unclear, and the related experiments do not adequately support the authors' claims. Furthermore, the authors have not articulated the motivation for using a diffusion model. This paper appears to simply combine LLM, trajectory encoding, and diffusion model together, when in essence it is still training a class-free guided diffusion policy.

---

### Official Review · Reviewer_cfeX · 2025-10-21

**Soundness:** 2
**Presentation:** 3
**Contribution:** 2
**Rating:** 4
**Confidence:** 4

**Summary:**

This paper proposes LLMDPD, in which text prompts and trajectory prompts are respectively encoded as conditions for a context-aware prompt-conditioned diffusion-based policy, with the aim of enhancing the capability of generalization in unseen tasks. The experiments on Meta-World and D4RL benchmarks demonstrate the effectiveness and generalization capability of the proposed method, outperforming various domain-relevant baselines.

**Strengths:**

* This paper introduces a conceptually clear and straightforward framework with relevant motivation. The paper writing is clear and easy to understand.
* The experimental results show strong and solid advantage compared with multiple baselines, demonstrating the effectiveness of LLMDPD.

**Weaknesses:**

* The method is quite intuitive, and the novelty is quite marginal. Simply combining prompt encoders with policies can be quite common in this field (e.g., Prompt-DT).
* Some of the evaluation tasks in the experiments are quite simple. Halfcheetah, hopper, and walker2d have relatively low dimensions in states and action spaces, making these tasks too simple to serve as a testbed to verify the effectiveness of such a proposed framework.

**Questions:**

* How are the unseen tasks defined in terms of D4RL experiments? At line 415, it is said "We evaluate the ability of the offline RL agent to generalize to unseen states and actions compared to the medium-replay offline datasets." Does this mean each state and action that did not appear in the medium-replay offline datasets can be viewed as an initial of an unseen task? (This might be less consistent with the general understanding of "unseen tasks", which usually differ in goals and termination conditions.)
* In the experiments of this paper, what are the implementation details of the proposed method and baselines (since no appendices or supplementary materials are provided)? For some of the baselines, the original implementations are simple multi-layer perceptrons (MLPs). However, LLMDPD, as a framework with transformer-based encoders and diffusion-based policies, possibly has a much more complicated architecture with many more parameters. In this way, we may not be able to judge whether the performance gain and generalization abilities come from the idea of text-and trajectory-based prompts with diffusions, or do they owe to a more complicated architecture with more parameters.
* The scalability of the proposed method. The experiments in this paper are conducted on state-based tasks with privileged information, with relatively low dimensions in state spaces. How can such a method be extended to environments with more complex inputs, such as visual-based tasks with image inputs? There may exist some problems, including but not limited to that encoding a trajectory of image-based observations and actions can be difficult and time-consuming.

---

### Official Review · Reviewer_fiay · 2025-10-23

**Soundness:** 2
**Presentation:** 3
**Contribution:** 1
**Rating:** 2
**Confidence:** 4

**Summary:**

This paper addresses the challenge of generalization to new tasks or environments in reinforcement learning. The authors propose LLM-Driven Policy Diffusion (LLMDPD), a framework that leverages large language models (LLMs) to enhance policy generalization. LLMDPD utilizes text-based task descriptions and trajectory prompts as conditional inputs to a context-aware policy-level diffusion model, enabling the agent to generalize more effectively to previously unseen tasks. Experiments on unseen-task benchmarks demonstrate that LLMDPD improves generalization and adaptability across diverse environments.

**Strengths:**

1.	The idea of leveraging large language models to encode task-level context for guiding offline policy learning is novel and inspiring, offering a promising direction for enhancing generalization in RL.
2.	The paper is well-organized and clearly written, making it easy to follow the motivation, methodology, and experimental results.

**Weaknesses:**

1.	The paper conflates two distinct forms of generalization: (i) out-of-distribution generalization within a single task, and (ii) generalization to unseen tasks. It remains unclear how LLMDPD addresses both types simultaneously, or whether improvements in one domain necessarily translate to the other. A clearer conceptual distinction and empirical validation of these two generalization aspects are needed.

2.	The experimental setting is non-persuasive:

    a)	    The Meta-World experiments aim to evaluate generalization to unseen tasks, yet the selected baselines are somewhat outdated—the most recent being MTDIFF (2023). Moreover, these baselines are primarily designed for multi-task learning rather than unseen-task generalization. Additionally, LLMDPD leverages task descriptions and prompts during inference, effectively granting task-awareness not available to competing baselines, which raises concerns about fairness in comparison.

    b)	    The D4RL experiments seem misaligned with the stated motivation of unseen-task generalization. Furthermore, since Diffusion-QL already introduced diffusion-based policy modeling, it is unclear whether the reported improvements stem solely from the inclusion of text and trajectory prompts or something others. This discrepancy also suggests an unfair advantage, as LLMDPD accesses contextual information unavailable to Diffusion-QL.

    c)	    The use of a pretrained LLM and transformer-based prompt encoder introduces additional parameters and computational capacity, making comparisons with non-LLM baselines potentially inequitable.

3.	Since the principal novelty of LLMDPD lies in incorporating textual and trajectory prompts, the ablation analysis should more thoroughly examine their respective contributions. What kinds of text prompts yield the greatest improvement? How sensitive is performance to prompt quality or ambiguity? Without such analyses, the generalizability of LLMDPD to other domains or less well-defined tasks remains uncertain.

**Questions:**

See the weakness.

---

### Official Review · Reviewer_Bwkm · 2025-11-02

**Soundness:** 2
**Presentation:** 2
**Contribution:** 2
**Rating:** 2
**Confidence:** 4

**Summary:**

This paper presents LLM-Driven Policy Diffusion (LLMDPD) which is an innovative framework aimed at improving the generalization ability of offline reinforcement learning (RL) by integrating textual knowledge and trajectory-based context. The approach leverages large language models (LLMs) to transform task descriptions to provide prior knowledge about the task. In addition, trajectory prompts consisting of states and actions are used to supply task-specific information. The policy is represented as a diffusion model conditioned on both the text embedding and the trajectory prompt. LLMDPD is trained using a combination of behavior cloning and policy improvement losses, similar to Diffusion-QL. Empirical results on the Meta-World and D4RL benchmarks show that LLMDPD outperforms existing methods in terms of generalization performance.

**Strengths:**

- The paper is clear and well-structured in presenting the methodology and experiments.
- The idea of using text embeddings from LLMs to provide prior knowledge for offline RL is novel and interesting.

**Weaknesses:**

- The paper does not provide a rigorous or unified definition of “generalization ability.” It mixes two types of generalization in Meta-RL and out-of-distribution generalization in offline RL.
- The motivation for leveraging these combined modules (LLMs and Diffusion Policy) is not well justified. The paper seems to be a minor extension of Diffusion-QL with text embeddings and trajectory prompts.
- This paper lacks a comprehensive comparison with existing methods in offline RL and Meta-RL. The baselines used are not appropriate for the tasks considered.

**Questions:**

- Could you formally define the “generalization ability” discussed in this paper and justify why Meta-World and D4RL are suitable benchmarks for measuring it?

- How does LLMDPD fundamentally differ from previous LLM-based approaches for offline RL or Meta-RL ([1][2][3]) beyond conditioning details?

- What is the rationale for choosing OLMo-1B in the ablation study, and can you provide additional results using alternative language models such as Qwen or LLaMA?

- What are the hyperparameters?


[1] Prompting Decision Transformer for Few-Shot Policy Generalization, 2022

[2] Towards an Information Theoretic Framework of Context-Based Offline Meta-Reinforcement Learning, 2024

[3] Meta-DT: Offline Meta-RL as Conditional Sequence Modeling with World Model Disentanglement, 2024

---

### Meta-Review · Area_Chair_FcpV · 2026-01-07

**Summary:**

Main concerns include:
* Multiple reviewers noted that LLMDPD appears to be a minor extension of Diffusion-QL, simply adding text and trajectory prompts. The combination of LLMs with prompt-conditioned policies is considered intuitive but not novel (e.g., Prompt-DT).
* Reviewers unanimously criticized the absence of a formal, unified definition of "generalization ability." The paper conflates two distinct concepts: (i) meta-RL generalization to unseen tasks and (ii) out-of-distribution generalization within a single task in offline RL, without clarifying how the method addresses both or whether improvements transfer between them.
* Uses outdated baselines (e.g., MTDIFF from 2023) designed for multi-task learning, not unseen-task generalization.
* LLMDPD uses task descriptions and trajectory prompts at inference time, giving it task-aware information unavailable to competing baselines, yet performance gains are not clearly disentangled from this advantage.

The reviewers found the paper well-written but methodologically weak. The core idea is intuitively appealing, but it suffers from conceptual confusion, questionable experimental design, insufficient novelty, and unfair baseline comparisons. The lack of rigorous ablation, missing implementation details, and scalability concerns further undermined confidence in the contribution. These systemic issues led to a consensus rejection despite acknowledging the clarity of presentation.

**Reviewer Concerns:**

No rebuttals wre provided.

**Reviewer Scores:**

All the reviewers agree the paper should not be accepted.

---

### Decision · Program_Chairs · 2026-01-26

Reject